# Parametrical Function Describing Influences of the Redistribution of Incorporated Oil for Rupture Process Reconstruction in Rubber

**DOI:** 10.3390/polym15061363

**Published:** 2023-03-09

**Authors:** Sanjoy Datta, Radek Stoček, Evghenii Harea

**Affiliations:** Centre of Polymer Systems, Tomas Bata University in Zlín, tř. Tomáše Bati 5678, 760 01 Zlín, Czech Republic

**Keywords:** rubber, fractography, tearing energy, oil concentration, deformation speed, infrared spectroscopy, parametrical functions

## Abstract

The present work is focused on finding (i) the tearing energy at rupture and (ii) the redistribution of incorporated paraffin oil on the ruptured surfaces as functions of (a) the initial oil concentration and (b) the speed of deformation to the total rupture in a uniaxially induced deformation to rupture on an initially homogeneously oil incorporated styrene butadiene rubber (SBR) matrix. The aim is to understand the deforming speed of the rupture by calculating the concentration of the redistributed oil after rupture using infrared (IR) spectroscopy in an advanced continuation of a previously published work. The redistribution of the oil after tensile rupture for samples that have three different initial oil concentrations with a control sample that has no initial oil has been studied at three defined deformation speeds of rupture along with a cryo-ruptured sample. Single-edge notched tensile (SENT) specimens were used in the study. Parametric fittings of data at different deformation speeds were used to relate the concentration of the initial oil against the concentration of the redistributed oil. The novelty of this work is in the use of a simple IR spectroscopic method to reconstruct a fractographic process to rupture in relation to the speed of the deformation to rupture.

## 1. Introduction

The statement that “the present work is an advanced continuation of a previous work” could have been told at some stage later, but it does not sound out of place when stated in the beginning. Rather, it highlights the whole theme behind this work.

In the previously published work [1], notched cured rubber tensile specimens, initially incorporated with specified concentrations of non-reacting process oil, were subjected to programmed tensile ruptures. Concentrations of the redistributed oil after a rupture on the ruptured surfaces as a function of speed for the tensile deformation were calculated using infrared spectroscopy. These redistributed concentrations were further related to the total energy of rupture (termed the tearing energy). A trend was found in the relation between the deformation speed and the concentration of the redistributed oil. Studying this trend, the reverse approach was imagined to be of much more practical importance.

In the reverse approach, a comparison between the concentration of the redistributed oil on the ruptured surfaces against an unruptured one was thought of. It was believed that through such a comparison, the speed of the deformation to such a rupture could be assessed, thus paving a novel way to understand a fractographic process in terms of reconstructing the tensile speed to failure through the rupture. This paved the scope of the present work.

In this present work, instead of only two studied tensile deformation speeds at only two initial oil concentrations, as was accomplished in the previous work [1], another higher speed to rupture along with a cryo-ruptured sample was introduced. For each of the defined speeds, the redistribution of oil after rupture was studied at four different initial oil concentrations (0.0, 4,5, 9.0, and 13.5 phr), which was structured to relate the deformation speed causing rupture with the redistributed oil concentration after rupture through the parametric fitting of the obtained data in the equations.

A chemically non-reactive liquid [2], such as a processing oil, initially homogeneously incorporated in a cured rubber matrix with a pre-existing crack, migrates in the volume of the rubber, which is dependent on the stress concentration caused due to the deformation of the rubber matrix under loading. This, in general, is dependent on the viscoelastic property of the rubber matrix, the viscous property of the liquid, the loading condition of the whole material, and the respective stress concentration in the vicinity of the crack tip following an energy dissipation mechanism [3]. In the present work, the chemical non-reactivity of a paraffinic processing oil with styrene butadiene rubber (SBR) and the subsequent redistribution of the oil at tensile rupture were used advantageously to support the objective of the present work.

However, in terms of miscibility, paraffinic oil (non-aromatic) is not a good choice for SBR (containing aromatic styrene units), and an aromatic oil would have been better. Still, there was a reason behind choosing paraffinic oil. It was required to obtain the separate characteristic infrared peak heights of the oil and the rubber. The characteristic peak height of SBR at 699 cm^−1^ was due to the presence of the styrene unit. This peak height was normalised to one to find the concentrations of the paraffinic oil with a characteristic peak height of 2915 cm^−1^. Had an aromatic oil been chosen, then, though the miscibility with SBR would have been better, the aromaticity of the oil would have produced a response at the same characteristic peak height of SBR, and thus, in effect, not allowing the normalisation of this peak height for SBR. Thus, in order to obtain unique characteristic peak heights of the rubber and oil, this combination was chosen. Thereafter, much care was taken to carry out the mixing, moulding, and further experiments with tensile rupture and IR studies very fast. This fast operation arrested the diffusion of the oil with time, if at all, allowing it to study only the redistribution of the oil after rupture.

Related to the choice of elastomer, natural rubber (NR) was excluded. NR is a material where strain-induced crystallisation (SIC) occurs, which comes into account at high deformations and is also dependent on the speed of the deformation [4]. Further, it is dependent on carbon black (CB) filler loading as well. It was shown that SIC occurs firstly at a strain of about 200% for 20% by volume that is CB filled and at about 400% for unfilled NR [5].

Thus, a non-SIC rubber, SBR [6,7], was preferentially used in the present study, leaving space for studying NR in some future works. This was targeted to investigate the influence of the redistribution of initially homogeneously and incorporated oil at rupture, after experiencing a large deformation to such a rupture, as a function of the variously increasing deformation speed to rupture, without further complicating the study with the additional effect of SIC.

## 2. Materials and Methods

The SBR of grade Kralex 1500, Synthos SA, USA, was mixed with paraffin oil–Tudalen 3912B, H&R Group, Hamburg, Germany. Other compounding ingredients such as zinc oxide, stearic acid, carbon black HAF N-330, rubber accelerator CBS, and sulphur, procured from local suppliers, were mixed in an internal mixer (Brabender Plastograph, Brabender GmbH & Co. KG, Duisburg, Germany) at a rotor speed of 50 rpm and at a temperature of about 60 °C. The mixing was accomplished at a fill factor of 0.85 for the mixer void volume of 50 cm^3^. The mixed batches were cured at 150 °C at a pressure of 200 kPa in a hydraulic press (LabEcon 300, Fontijne Presses, Delft, The Netherlands) at the optimum cure time, t_90_. = 13.5 min, which was determined in a moving die rheometer (MonTech MDR 3000 Basic, GmbH, Buchen, Germany). The compositions of the four prepared batches with the initial oil concentrations of 4.5, 9.0, 13.5, and 0.0 phr are shown in Table 1. The first three batches were used for the main experimental studies on the redistribution of the paraffin oil, and the fourth one, which had no paraffin oil, was used as the control.

The compression of moulded samples with the dimension 20 mm × 10 mm × 2 mm (length, L_0_ × width, Q × thickness, B) and with cylindrical shoulders of a 3mm radius (R3) were notched by an initial length a = 1 mm from the edge to obtain single edge notch tensile (SENT) specimens (Figure 1) Customized equipment with a metal blade was used to notch the samples. The dimension of the various test specimens in relation to similar studies can be found in detail in some specific works [8,9].

Sample designation: The various oil-incorporated samples were designated as SBR_x, where x = the initial oil concentration in phr; for example, SBR_4.5 indicates that the SBR rubber sample was incorporated with 4.5 phr of oil.

The prepared samples were subjected to the following experimental programmes.

The static deformation of the samples: the notched samples incorporated with initial oil were deformed to 300%, 600%, and 900% of their original lengths, and the deforming crossheads were arrested in the static condition for each. The deformation was effected by a hand vice device in which the speed of the deformation to achieve the statically deformed condition was not quantified. The aim was to find the oil concentrations at various places on the surface of the deformed samples.

The tensile and cryo-rupturing of the samples: each notched sample was fixed in a clamping system by using cylindrical shoulders of radius R3 (Figure 1) on both sides to avoid any slippage of the sample from the clamps during a uniaxial tensile test. For each sample, the tensile testing protocol was carried out at three different deformation speeds of 10, 100, and 500 mm∙min^−1^ in a universal tensile testing machine (Testometric, Rochdale, UK) to effect elongation followed by tensile rupture. Three repetitions from each of the four compounded and cured rubber compositions (0.0, 4.5, 9.0, and 13.5 phr) were analysed per each deformation speed.

Another rupture protocol was effected by cryo-rupturing the variously compounded samples by submerging them inside liquid nitrogen, bringing them in temperature equilibrium with that of the temperature of the liquid nitrogen (−195.7 °C) [10] and finally applying pressure to the rupture of the samples with the generation of new surfaces.

During this tear-rupture process, the physical area generated on each of the two surfaces of the sample was A = (Q-a)B. The dimensions are shown in Figure 1.

Calculation of tearing energy: The tearing energy (henceforth called the total energy to rupture), T, was calculated by using the absolute area type integration in OriginPro 8.5 [11]. This is the area between the curve and the deformation axis in the force–F versus deformation–l plot, starting from (0, 0) to the deformation at rupture. The values of F and corresponding l were obtained from the non-loading state up to the deformation at rupture; the deformation observed in the direction of uniaxial tensile loading was parallel to the length L_0_ of the sample.

In the present work, the calculated energy was the total energy to rupture and was not the absolute value of the tearing energy. However, since the energy to rupture amongst the different samples was more important and served the specific purpose of this work, this energy was termed the tearing energy. A detailed calculation of tearing energy using the J integral is to be found in many articles [8,12,13,14,15].

Calculation of redistributed oil concentration using attenuated total reflection Fourier transform infrared (ATR FT-IR) spectroscopy:

Firstly, it was important to understand the concentration of the redistributed oil at different locations on the exposed surfaces of the samples and under static deformation before rupture. This phenomenon was assumed to have an effect on the final redistributed oil concentration after the rupture in dynamic conditions.

The sample incorporated with 9.0 phr of initial oil was deformed to 300%, 600%, and 900% of its original length, respectively, at a time and was tested for this purpose. The redistributed oil concentrations in the deformed states were determined at three places on the deformed surface–(1) 1 mm behind the deformed crack tip–designated as “front”, (2) a far-away defined place very near to the grip of tensile deformation designated as “far” and (3) a place 1 mm above the notch designated as “side”. The places are shown in Figure 2 [1]. The tests were performed immediately after the deformed lengths were attained.

This experiment was designed based on the following assumption: since the digital image correlation (DIC) yielded a strain distribution when viewed on the surface of the test specimen [1] at a static strain, it was assumed that the oil concentration might also show a distribution pattern on the surface of (L_0_ × Q) in the static strain. The uniaxial deformation along the length of the sample also caused deformation in the other two perpendicular directions, the monitoring of which was not performed using the DIC technique. Thus, the study of the distribution of the oil concentration under static deformation was limited only to the surface (L_0_ × Q) but was helpful in explaining what happened later after the total rupture.

The values at only 9.0 phr were reported in this work to avoid the unnecessary lengthening of the manuscript. The tests were also conducted at various other concentrations, where very similar trends were observed.

Secondly, and more importantly pertaining to the present work, the redistributed oil concentration in a dynamic deformation to rupture on the ruptured surfaces was calculated. ATR FT-IR studies of surfaces in a static deformation as well as the newly created ruptured surfaces after the uniaxial tensile rupture of the specimens were conducted in a wavenumber range from 4000 to 650 cm^−1^ using an infrared spectrometer (Thermo Scientific Nicolet iS5, Waltham, MA, USA). The spectra were obtained at a resolution of 2 cm^−1^ using a germanium crystal.

The volume of the test under IR studies was the volume generated by the evanescent beam inside the rubber material. The evanescent beam consisted of a base diameter and a height = the depth of penetration of the beam, *d_p_* inside the rubber. In general, *d_p_* is a function of the refractive indices, *μ*_1_ and *μ*_2_ of the ATR FT-IR crystal and the sample, respectively, alongside the wavelength, *λ* and the angle of incidence, *θ* of radiation. Though a detailed calculation of the parameters affecting the volume has not been shown here, it can be supported that this volume of the test was small enough to register the surface concentration of the oil but was large enough for an otherwise homogeneous rubber matrix.

The calculation that was used to find the redistributed oil concentration under static tensile deformation was the same as those that were applied for the tests under a dynamic tensile deformation to rupture.

In the dynamic tensile deformation to rupture, the newly generated and ruptured surfaces were subjected to ATR FT-IR measurements five minutes after tensile rupture. This quick interval of five minutes was to arrest the diffusion of oil if any.

The characteristic absorbance peak height of the oil at 2915 cm^−1^ [16] was determined at a distance, a_s_ = 3 mm, i.e., 2 mm behind the initial notch of 1 mm from the edge of the sample (Figure 1), [1] on the two newly generated surfaces. For the three repetitions for each of the samples, at each deformation speed, the arithmetic means concentration from the two generated ruptured surfaces was reported as the redistributed oil concentration. To quantify the peak heights, the spectrum for each sample was modified with an algorithm of baseline subtraction [17,18,19].

The procedure that was used for the quantification of the characteristic peak for oil at 2915 cm^−1^ is now discussed in detail. The peak at 2915 cm^−1^ was also obtained for the sample at 0 phr of oil, and it was attributed to the -CH_2_ bond vibration from SBR [20]. This peak height at 0 phr of the initial oil concentration was quantified against the characteristic peak height of SBR at 699 cm^−1^ [21], with the latter normalized to a value of one. It was found that the peak height at 2915 cm^−1^ at 0.0 phr of the initial oil concentration was independent of the deformation speeds of 10, 100, and 500 mm·min^−1^ and was also independent of the cryo-ruptured programme, with an average value of 0.2575 a. u., and a standard deviation of 0.0160. This average value was due to SBR only and was thus always subtracted from unsubtracted peak heights for the redistributed oil concentrations at the same wavenumber of 2915 cm^−1^ and for all the other tensile ruptured programmes. For the oil-incorporated samples, the unsubtracted values of the peak height were obtained against the normalized peak height of SBR at 699 cm^−1^. The subtracted values were the final values of the peak heights of the redistributed oil after rupture for the variously initially oil-incorporated samples.

For each prepared batch, the reported result of the IR peak height was the median value of the three calculated values which varied only within a negligible limit.

## 3. Results and Discussion

The fate of the initial oil concentration as a redistributed oil concentration after rupture was definitely a reflection of what was initiated with the deformation before the rupture. Thus, static deformation tests were conducted prior to the dynamic deformation. This was ascertained by a pattern in the redistribution as observed under defined static strains. In the present work, it has been demonstrated with the sample and an initial oil concentration of 9.0 phr. The three different static deformation states and the positions on the surface of the deformed samples where the redistributed oil concentration was measured have been stated in the experimental section in detail. The values obtained are shown in Table 2.

For all the three statically deformed elongations, the concentration of the redistributed oil followed the decreasing pattern of side > far > front. This implies that the oil was squeezed out in front of the notch during the deformation process.

Having found this distribution pattern in static deformation, the discussion then continues with the results obtained from the dynamic deformation to rupture. The progress of the tensile deformation from zero to rupture at a very high deformation length signified that the sample kept on deforming in a uniaxial deformation under a constant deformation speed with only notch blunting and no crack propagation. At a moment nearing the end of the deformation process, a crack was formed behind the notch which suddenly propagated very fast, perpendicular to the direction of the crosshead displacement, to cause rupture. The notch blunting can be visualized from Figure 2. Additionally, in a previous work by some of the authors of the present work, and very similar to the present work, the phenomenon of notch blunting was represented in figure [1].

A representative plot of force versus the deformation of the sample SBR_9.0 at a deformation speed of 500 mm·min^−1^ is shown in Figure 3. The absolute area integration of this plot generated the value of the tearing energy. The calculation of the tearing energy is discussed in detail in the experimental section. The plots for all other tensile ruptured samples at various deformation speeds and at various initial oil concentrations produced similar curves and are not shown here to restrict the volume of the article within a limit.

This discussion was further continued with the results obtained from the IR studies. Here, the results from all the programmes are shown as the work was focused more on the IR technique to find the initial and also on the redistributed concentration of the oil. Figure 4, divided into four parts (a), (b), (c), and (d), according to the four initial oil concentrations of 0.0, 4.5, 9.0, and 13.5 phr, respectively, presents the baseline subtracted IR spectra for SBR at 699 cm^−1^ and for the redistributed paraffin oil at 2915 cm^−1^ after rupture [17,18,19]. In each sub-part of the figure, only one spectrum is shown at random from a repetition of three (as stated in the experimental part) for each of the following rupturing programmes– at), (I)-10, (II)-100, and (III)-500 mm·mim^−1^ (tensile ruptured) and (IV)- 0 mm·min^−1^ (cryo-ruptured).

Only one representative plot has been shown for each unique programme to avoid the overcrowding of figures, keeping in mind that in a repetition of three, the results were very similar.

The calculated values for the tearing energies for the various oil-incorporated samples and the corresponding calculated values of the IR absorbance peak heights at 2915 cm^−1^ (representing the redistributed oil concentration after rupture) are shown in Table 3.

The results presented in Table 3 are summarized in Figure 5, Figure 6 and Figure 7 for quick visualization of the following, respectively: (a) tearing energy as a functions of deformation speed, (b) concentration of the redistributed oil as a function of deformation speed, and (c) concentration of the redistributed oil as a function of tearing energy.

The principal observed facts from the figures with explanations are as follows:

(a) At any defined initial oil concentration, the tearing energy can increase with a decrease in the deformation speed, the minimum being 500, with an intermediate value at 100 and a maximum at 10 mm∙min^−1^ (Figure 5).

Explanation–This was because more energy was being used for a higher elongation at a rupture with lower speeds, as has already been explained in [1]. Additionally, the deformation was more viscoelastic at the lowest deformation speed, and much energy during the deformation process contributory towards viscous loss along with the energy for the elastic deformation. Thus, more total energy was required for the deformation.

(b) At any defined initial oil concentration, the IR absorbance peak heights for the redistributed oil at 2915 cm^−1^ were very similar for the cryo-ruptured, and also for the 10 and 500 mm·min^−1^ tensile ruptured samples but were significantly lower for the 100 mm·min^−1^ tensile ruptured sample as is very well captured in Figure 6.

Explanation–The cryo-rupturing of the samples was enhanced at a low temperature of −195.7 °C (temperature of liquid nitrogen). Since the samples were broken instantaneously on the application of pressure, the oil did not have time to become redistributed. Thus, the condition was a measure of the initially homogenously mixed oil concentration.

At 500 mm·min^−1^, the total time taken for any of the samples to deform to the maximum elongation followed by the immediate rupture was very little. As a consequence, the viscous oil did not receive much time for redistribution near the position immediately behind the notch where the experiment was focused. This was also the position where the maximum stress was developed. Thus, the concentration of the oil after rupture, was almost the same as that of the concentration of the cryo-ruptured (initial concentration) samples with a marginally larger amount.

At 10 mm·min^−1^, the deformation speed was so slow that at every step of the deformation there was ample time for the oil to move into the concentration equilibrium with that of the initial concentration. This was reflected in the experimental results, which depicted almost the same oil concentration as was observed for the cryo-ruptured samples.

At an intermediate speed of 100 mm·min^−1^ after rupture, the generated surfaces contained fewer amounts of oil compared to that of the initial concentration and also compared to the concentrations on the ruptured surfaces of all other tensile deformation programmes. Thus, a greater portion of the redistributed oil must have gone somewhere inside the rubber matrix if it did not evaporate. However, the present work focused only on the redistributed oil concentration on the ruptured surfaces to understand the speed of the rupture. Thus, further investigation on the redistribution of the oil in other parts of the rubber matrix was not conducted.

It was assumed that, in the 100 mm·min^−1^ programme, the rubber was ruptured before the oil regained an equilibrium concentration. This was, reflected through a smaller amount of oil on the ruptured surfaces.

(c) The plot of IR absorbance peak height(proportional to the redistributed oil concentration after rupture) versus the tearing energy(proportional to the lowering of the deformation speed) (Figure 7) showed an interesting trend for all the samples. It shows that for any oil concentration, though the tearing energy kept on increasing with a decrease in the deformation speed, the redistributed oil concentration passed through a minimum at 100 mm·min^−1^. The redistributed oil concentration for the 10 and 500 mm·min^−1^ was similar. This supports the explanation of the previous section (b).

A further explanation is as follows: At 500 mm·min^−1^, the stored elastic energy before the rupture made the rubber primarily deform in an elastic manner and made the rubber and the oil return back to the ruptured surface in a similar proportion as they were present initially. This was reflected as a similar oil concentration was detected in the cryo-ruptured samples.

At 10 mm·min^−1^, the deformation speed was very slow, and there was ample time for the rubber to deform in a more viscous manner over the 500 mm·min^−1^ deformation. This made a smaller amount of rubber matter to regain the original mass at the position on the ruptured surfaces. Additionally, the less elastically stored energy in the rubber matrix forced back less oil which was squeezed out during the deformation process. The proportions of the regained masses of the rubber and the oil were such that they were similar to those of the cryo-ruptured and the 500 mm·min^−1^ tensile ruptured samples.

That the oil was squeezed out is supported by the pattern of the redistributed oil as was observed during the static deformation programmes, which have already been explained at the beginning of this section. The squeezing out was more pronounced for the 10 mm·min^−1^ deformation speed. At this deformation speed, a permanent set was visually observed (not quantified in the present work) just after rupture. This set was much lower at 100 mm·min^−1^ and was visually non-detectable at 500 mm·min^−1^.

Finally, the 100 mm·min^−1^ programme allowed the rubber to regain the mass on the deformed surface in a mostly elastic manner, but the elastic stored energy was not significant to force back much of the squeezed-out oil. Thus, the redistributed concentration was the lowest.

(d) For each deformation programme, the ratio among the three initial oil concentrations (4.5:9.0:13.5) was also approximately the ratio amongst the redistributed oil concentrations after the rupture, as observed in Table 3. This supported the correctness of choosing the characteristic IR wavenumber of 2915 cm^−1^ for the present study.

Data from Table 3 were used to plot Figure 8, relating the concentration of the initial oil against the concentration of the redistributed oil. However, this was limited to only four deformation speeds which were experimentally undertaken.

In order to understand the relation at shorter and more regular intervals over the entire deformation speed from 0–500 mm·min^−1^, the following two steps were further undertaken:(a)From the dataset used for plotting Figure 8 and Figure 9. was generated. Figure 9 shows parametrically fitted curves relating the redistributed oil concentration as a function of the deformation speed for each of the initial oil concentrations.(b)From the dataset obtained from the parametric fitting in Figure 9, selected ordered pairs of data at approximate intervals of 50 mm·min^−1^ were used to replot Figure 8 as Figure 10. Figure 10 shows the relation between initial oil concentration and redistributed oil concentration. Unlike Figure 8, it not only shows the relation at the experimental deformation speeds of 0, 10, 100, and 500 mm·min^−1^ but also at many more mathematically generated deformation speeds at 50 mm·min^−1^ intervals.

The fitted curves in Figure 9 generated Equation (1) which is as follows:(1)$$y=\mathrm{int}+B_{1}x^{1}+B_{2}x^{2}$$
where *x* = deformation speed in mm·min^−1^; *y* = IR absorbance (represents the concentration of the redistributed oil); int = intercept on the *y*-axis (represents the redistributed oil concentration at the cryo-ruptured condition); *B*_1_ and *B*_2_ are constants related to the first and the second power of *x*, respectively.

The various parameters for the parametric fitting of data in Figure 9, along with their statistics, are condensed in Table 4.

In Figure 10, for each of the deformation speeds, the data were fitted in the general equation of a straight line as shown in Equation (2).
(2)y=a+bx

*y* = the initial oil concentration

*x* = the redistributed oil concentration after rupture measured as the characteristic infrared absorbance

*a* = intercept on the *y*-axis as a measure of initial oil concentration when the redistributed oil concentration is zero

*b* = slope of the line and shows the change in initial oil concentration over a change in redistributed oil concentration

The fitted parameters of Figure 10 are presented in Table 5.

The regression squared values reflect a very good straight-line fitting for every deformation speed.

A trend in the redistribution of the oil after rupture was observed. The concentration of the redistributed oil at any of the initial oil concentrations of 4.5, 9.0, and 13.5 was in the decreasing order as follows: (500 mm·min^−1^ > 10 mm·min^−1^ tensile ruptured > cryo-ruptured) ≫>> 100 mm·min^−1^ tensile ruptured. Additionally, the cryo-ruptured as well as the 10 and 500 mm·min^−1^ tensile ruptured values were very close, as has already been discussed in detail. This means that the redistributed concentration decreased from a high deformation speed of 500 mm·min^−1^ to reach a minimum of 250 mm·min^−1^ and then again increased to reach the upper limit.

Of course, theoretically, had the process been very slow, as is the hypothetical case in a reversible thermodynamic process, the equilibrium at all stages during the elongation could have been realized. Under such a condition, there would have been no change in the concentration of the redistributed oil at any stage because then, at every small stage in the deformation, the oil concentration would have come into a concentration equilibrium with that of the initial oil concentration.

After having explained the parametric fitting, an explanation opposing the obtained fitting is also important. From the results of the present work, it was found that the concentration of the redistributed oil went through a minimum when plotted as a function of the experimental deformation speeds. The results from the experimentally obtained data housed that minimum at the deformation speed of 100 mm·min^−1^, whereas the fitted data reflected this minimum at a deformation speed of 250 mm·min^−1^.

The parabolic fitting was completely empirical, as there could have been other fittings as well. For instance, straight lines connecting the four experimentally obtained points could have been drawn. Either or neither of the two fittings could have been correct in asking for a third option.

However, the experimentally obtained data reflected a very important phenomenon. Th cryo-ruptured (representing the initial oil concentration) and the 500 tensile-ruptured samples (fast speed to rupture, representing more elastic deformation to rupture) showed very similar and higher redistributed oil concentrations, and the 100 mm·min^−1^ tensile ruptured produced a minimum. So, it was concluded that there must be a deformation speed between the two extremes where the concentration of the redistributed oil was at a minimum. Had experiments with more deformation speeds at regular intervals been conducted, then the minimum might have been at some other speed. However, there would have been a minimum.

Thus, this work has helped to draw a conclusion on the existence of a minimum in the relation between the speed of deformation (ranging from a very low to a very high speed) and the concentration of the redistributed oil after rupture, in notched, tensile ruptured styrene butadiene rubber.

## 4. Conclusions

Single edge notched tensile (SENT) samples from a non-strain induced crystallisation (SIC) cured styrene butadiene rubber (SBR) containing a non-reacting and infrared (IR) traceable processing paraffin oil at four initial concentrations of0.0, 4.5 9.0 and 13.0, were subjected to uniaxial tensile deformation programmes of 10, 100 and 500 mm∙min^−1^ deformation speeds to rupture. Another cryo-ruptured sample was also used in the work. The total energy required to elongate the sample in the uniaxial deformation direction and to crack propagate the notch on the sample to rupture was termed the tearing energy. The tearing energy as a function of deformation speed at all the defined oil initial concentrations was determined. Further, predefined areas on the generated ruptured surfaces were subjected to attenuated total reflection (ATR) Fourier transform infrared (FT-IR) spectroscopy to quantify the amount of redistributed oil on a defined position on the ruptured surfaces.

The observed trends in the concentration of the redistributed oil were explained in terms of the speed of deformation playing an important role to allow the oil to be redistributed. The redistributed concentration of the oil was explained through the simultaneous quenching of the oil and the viscoelastic deformation of the rubber matrix during the process of uniaxial elongation and their relative recovery after rupture.

It conclusively proved that the IR peak height response, representing the redistributed oil concentration after rupture was a function of the deformation speed and reflected a minimum at an intermediate deformation speed. This was supported through parametric fitting of the experimentally obtained data.

This conclusion is of immense importance in reverse engineering a rupture process in a compounded and cured rubber product. By measuring the concentration of oil on the newly formed surfaces after rupture, by using only a simple quantitative IR technique, the mechanical impact on the destroyed rubber product can be determined. This may be used for legal reconstruction of different accidents.

## Figures and Tables

**Figure 1 polymers-15-01363-f001:**
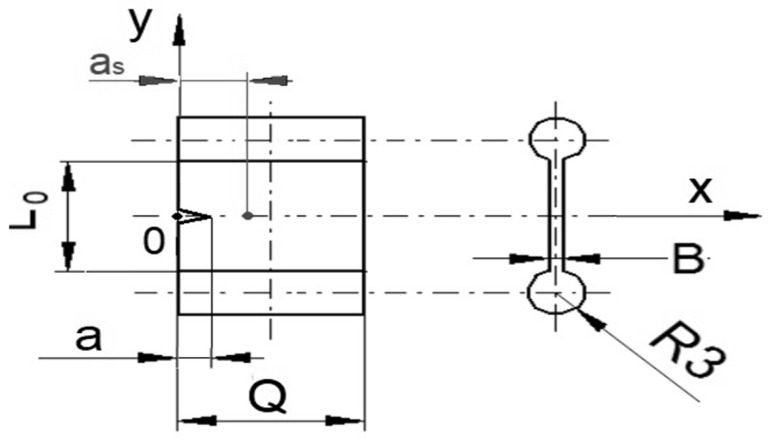
Geometry of single edge notched tensile (SENT) specimen.

**Figure 2 polymers-15-01363-f002:**
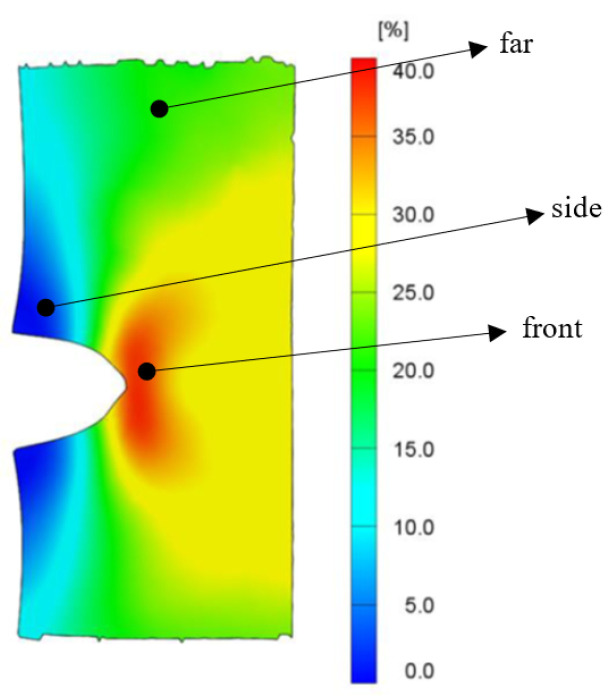
Positions of the three places where the concentration of the redistributed oil was measured under static strain.

**Figure 3 polymers-15-01363-f003:**
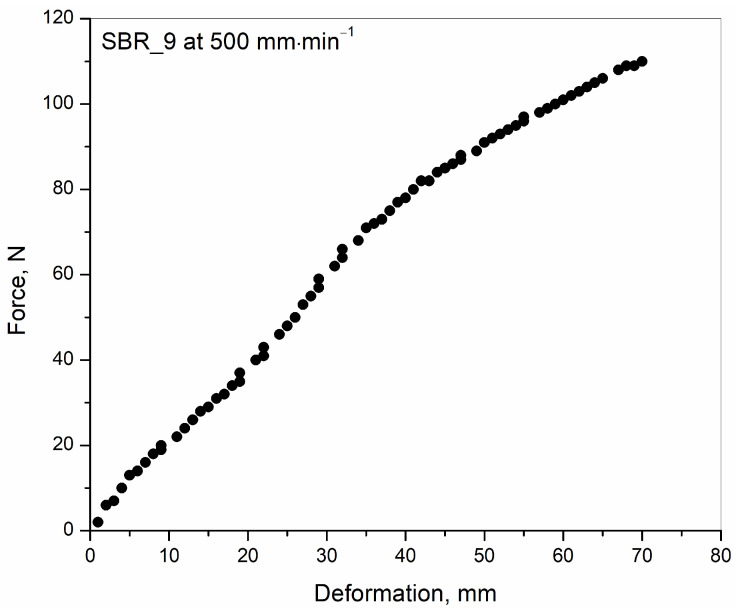
Plot of force versus deformation for the sample SBR_9.0 at a deformation speed of 500 mm, min^−1^ was used for the calculation of tearing energy.

**Figure 4 polymers-15-01363-f004:**
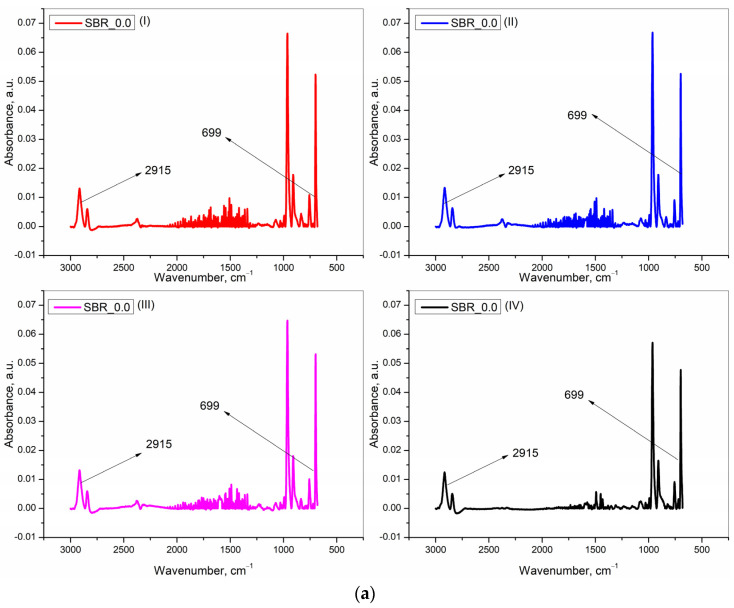
(**a**) Baseline subtracted IR spectra for the compound at 0.0 phr of the initial oil concentration, with absorbance peaks for SBR at 699 cm^−1^ and for a component other than oil at 2915 cm^−1^ for (I) 10, (II) 100, and (III) 500 mm·min^−1^ (tensile ruptured) and for (IV) cryo-ruptured samples. (**b**) Baseline subtracted IR spectra for the compound at 4.5 phr of the initial oil concentration, with absorbance peaks for SBR at 699 cm^−1^ and (a component other than oil + the redistributed oil) at 2915 cm^−1^ for (I) 10, (II) 100, and (III) 500 mm·min^−1^ (tensile ruptured) and for (IV) cryo-ruptured samples. (**c**) Baseline subtracted IR spectra for the compound at 9.0 phr of initial oil concentration, with absorbance peaks for SBR at 699 cm^−1^ and (a component other than oil + the redistributed oil) at 2915 cm^−1^ for (I) 10, (II) 100, and (III) 500 mm·min^−1^ (tensile ruptured) and for (IV) cryo-ruptured samples. (**d**) Baseline subtracted IR spectra for the compound at 13.5 phr of initial oil concentration, with absorbance peaks after baseline fitting and subsequent subtraction for SBR at 699 cm^−1^ and (a component other than oil + the redistributed oil) at 2915 cm^−1^ for (I) 10, (II) 100, and (III) 500 mm·min^−1^ (tensile ruptured) and for (IV) cryo-ruptured samples.

**Figure 5 polymers-15-01363-f005:**
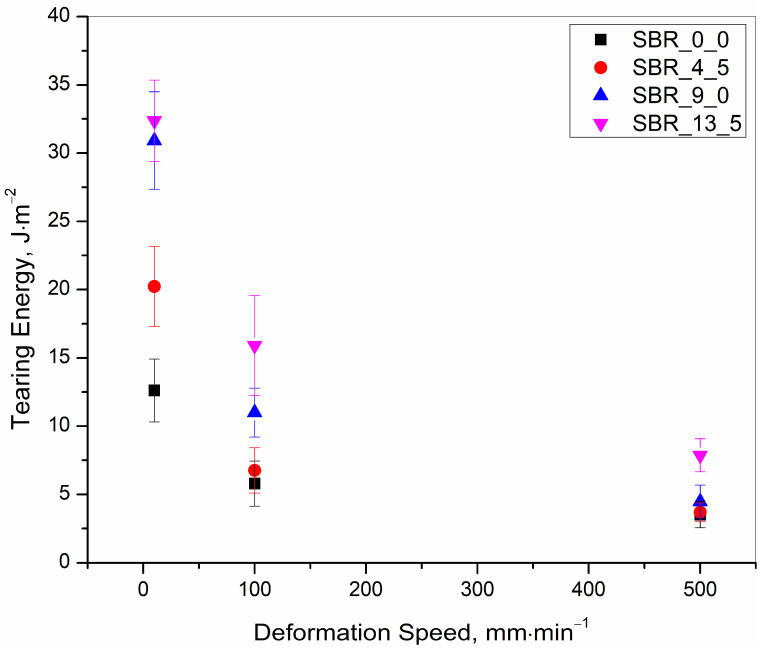
Tearing energy as a function of deformation speed for 0.0, 4.5, 9.0 and 13.5 phr of initial oil incorporated samples.

**Figure 6 polymers-15-01363-f006:**
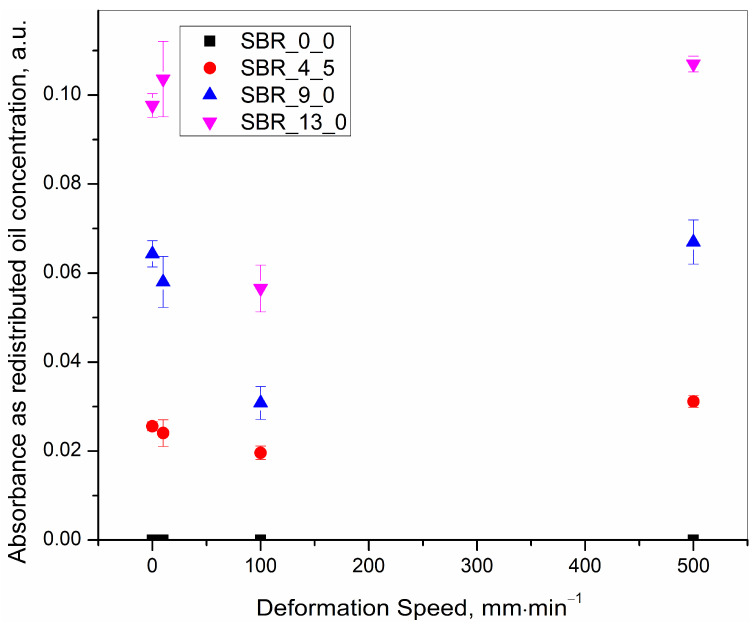
Calculated IR absorbance peak height at 2915 cm^−1^ representing the redistributed oil concentration after rupture as a function of deformation speed for 0.0, 4.5, 9.0 and 13.5 phr of initial oil incorporated samples.

**Figure 7 polymers-15-01363-f007:**
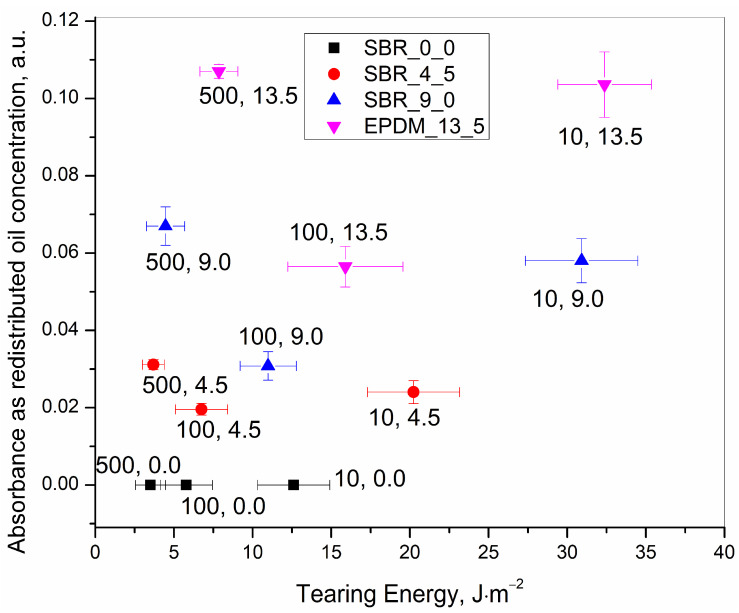
IR absorbance peak at 2915 cm^−1^ representing redistributed oil concentration after rupture as a function of calculated tearing energy at rupture for 0.0, 4.5, 9.0 and 13.5 phr of initial oil incorporated samples.

**Figure 8 polymers-15-01363-f008:**
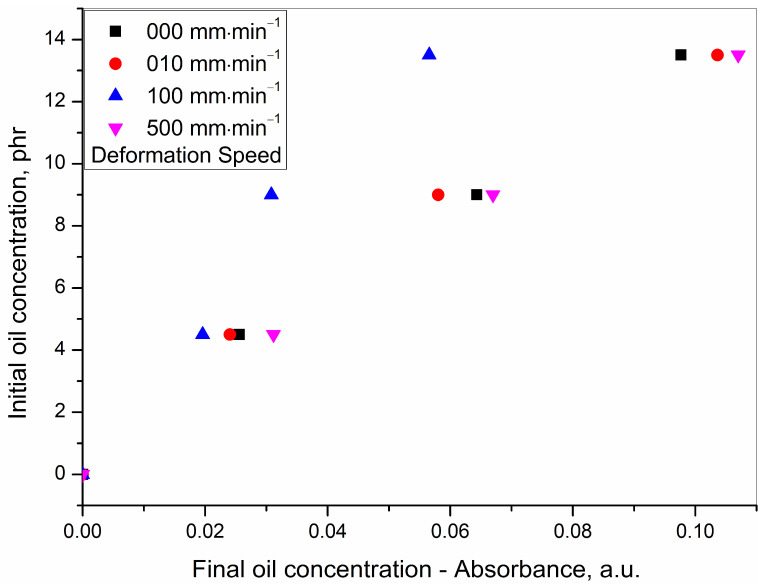
Initial oil concentration for various tensile ruptured programmes against redistributed oil concentration as obtained from experimental results.

**Figure 9 polymers-15-01363-f009:**
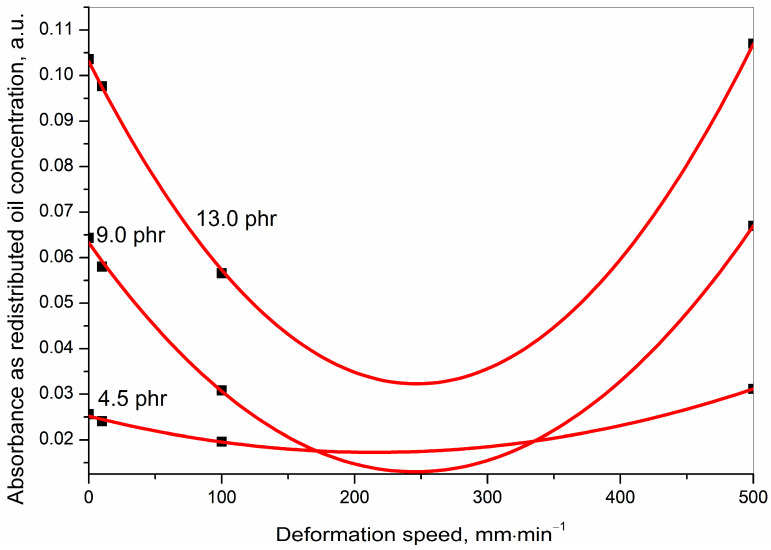
Redistributed oil concentration as a function of deformation speed for various initial oil concentrations.

**Figure 10 polymers-15-01363-f010:**
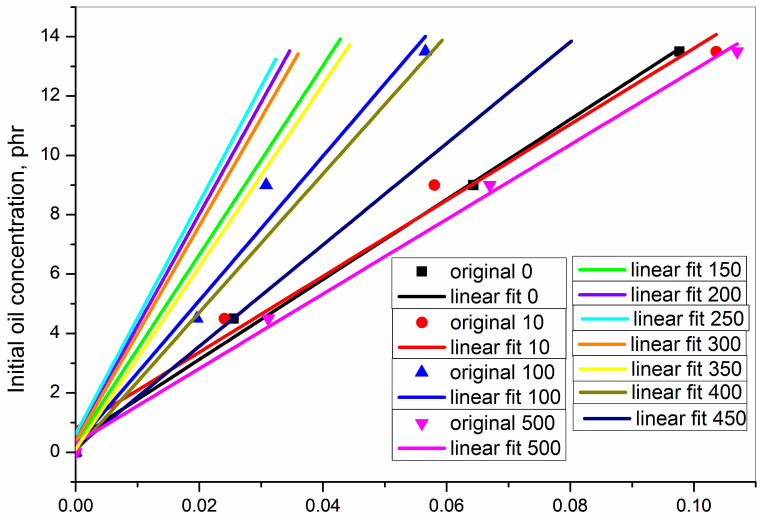
Initial oil concentration against redistributed oil concentration after rupture for various deformation speeds (the numbers in the legend represents the deformation speed in mm·min^−1^).

**Table 1 polymers-15-01363-t001:** Compositions of the various prepared batches.

Compounding Ingredients	phr *			
	Batch I	Batch II	Batch III	Batch–IV (Reference Batch)
SBR	100.00	100.00	100.00	100.00
Zinc oxide (ZnO)	3.00	3.00	3.00	3.00
Stearic acid	1.00	1.00	1.00	1.00
Carbon black, N-330	50.00	50.00	50.00	50.00
Paraffin Oil, Tudalen 3912B	4.50	9.00	13.50	0.00
Rubber accelerator, CBS	2.50	2.50	2.50	2.50
Sulphur	1.70	1.70	1.70	1.70

* Parts per hundred rubbers by mass.

**Table 2 polymers-15-01363-t002:** Redistributed oil concentration for various static deformation programmes.

Static Deformation, %	Sample Designation	Concentration of Redistributed Oil, IR Peak Height
300	SBR_9.0_far	0.036 ± 0.003 *
	SBR_9.0_front	0.007 ± 0.001
	SBR_9.0_side	0.091 ± 0.003
600	SBR_9.0_far	0.048 ± 0.002
	SBR_9.0_front	0.020 ± 0.001
	SBR_9.0_side	0.129 ± 0.005
900	SBR_9.0_far	0.069 ± 0.002
	SBR_9.0_front	0.045 ± 0.002
	SBR_9.0_side	0.115 ± 0.003

* The values after ± represents the standard deviation over an entire population of three readings.

**Table 3 polymers-15-01363-t003:** Tearing energy and the corresponding absorption peak height at 2915 cm^−1^ (representing the redistributed oil concentration after rupture).

Initial Oil Concentration, phr	0.0		4.5		9.0		13.5	
Deformation Speed, mm·min^−1^	Tearing Energy, J·m^−2^	Calc. Peak Height, a.u ^#^.	Tearing Energy, J·m^−2^	Calc. Peak Height, a.u.	Tearing Energy, J·m^−2^	Calc. Peak Height, a.u.	Tearing Energy, J·m^−2^	Calc. Peak Height, a.u.
0 (cryo-ruptured)	NA	0 ± 0	NA	0.026 ± 0.001	NA	0.064 ± 0.003	NA	0.098 ± 0.003
10 (tennsile-ruptured)	12.61 ± 2.30	0 ± 0	20.23 ± 2.94	0.024 ± 0.003	30.92 ± 3.58	0.058 ± 0.006	32.375 ± 2.97	0.103 ± 0.008
100 (tensile-ruptured)	5.79 ± 1.65	0 ± 0	6.75 ± 1.66	0.020 ± 0.002	10.99 ± 1.79	0.031 ± 0.003	15.901 ± 3.66	0.057 ± 0.005
500 (tensile-ruptured)	3.50 ± 0.95	0 ± 0	3.69 ± 0.69	0.031 ± 0.001	4.47 ± 1.22	0.067 ± 0.005	7.863 ± 1.21	0.107 ± 0.002

The values after ± represents the standard deviation over the entire population of three readings; ^#^ arbitrary unit.

**Table 4 polymers-15-01363-t004:** Parameters for the parametric fitting of curves in Figure 8.

Equation	*y* = Intercept + *B*_1_·*x*^1^ + *B*_2_·*x*^2^		
	Sample Designation		
Parameters of fit	SBR_4_5	SBR_9_0	SBR_13_5
Intercept	0.025	0.063	0.103
*B* _1_	−7.369 × 10^−5^	−4.093 × 10^−4^	−5.740 × 10^−4^
*B* _2_	1.712 × 10^−7^	8.334 × 10^−7^	1.164 × 10^−6^
Statistics of fitting			
Residual Sum of Squares	3.379 × 10^−7^	2.673 × 10^−6^	6.773 × 10^−5^
Adj. R-Square	0.985	0.990	0.876

**Table 5 polymers-15-01363-t005:** Parameters from the fitting of straight line relating the concentration of the initial oil versus the concentration of the redistributed oil.

Rupture Programme	Parameter *a*, Intercept	Parameter *b*, Slope	Regression Squared, R^2^
000, cryo-ruptured	0.61	134.73	0.99
010, tensile ruptured	0.57	128.49	0.99
050, tensile ruptured	0.44	175.49	0.99
100, tensile ruptured	0.30	243.43	0.99
150, tensile ruptured	0.29	318.42	0.96
200, tensile ruptured	0.49	376.43	0.92
250, tensile ruptured	0.61	390.40	0.90
300, tensile ruptured	0.37	362.75	0.92
350, tensile ruptured	0.11	307.16	0.95
400, tensile ruptured	0.44	233.44	0.99
450, tensile ruptured	0.15	170.94	0.99
500, tensile ruptured	0.30	125.76	0.99

## Data Availability

The data presented in this study are available on request from the corresponding author.

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
