# Peer review of "Parametrical Function Describing Influences of the Redistribution of Incorporated Oil for Rupture Process Reconstruction in Rubber"

_polymers, 2023, doi:10.3390/polym15061363_

Round 1

Reviewer 1 Report

Reviewer report on polymers-2147240

Parametrical Function Describing Influence of Redistribution of Incorporated Oil for Rupture Process Reconstruction in Rubber by Sanjoy Datta, Radek Stoček and Evghenii Harea.

This manuscript is focussing on results an experimental study on carbon black filled SBR compounds dealing with interrelations between deformation speed during fracturing, tearing energy and oil redistribution in the ruptured surfaces. It is claimed that existing correlations can used to get a simple measure (oil redistribution measured by IR spectroscopy) to determine deformation speeds without knowledge about fracturing conditions. SBR compounds with three different oil contents are investigated using four different fracturing conditions to confirm this hypothesis experimentally.

In general, it would be very interesting if the proposed approach will truly work. From this perspective the manuscript is interesting. On the other hand, the technical quality of the manuscript must be significantly improved. There are various points where the description of the performed experiments and the given interpretations need urgently clarification. The English is partly also needing revision. Major questions to be answered and central aspects to be clarified are described in more detail below together with a few minor points. From my point of view this manuscript must be substantially revised before a final decision can be formulated whether or not the work is suitable for publication in Polymers.

Major points

1.)       The authors tell in the experimental section that “FT-IR studies of the newly created fracture surfaces after uniaxial tensile rupture of the specimens at ambient temperature of 25 °C were done  …“. What does this mean? Is there a standard protocol used and how long does it take before the IR measurements are performed after the tensile tests? This can be important since diffusion processes are relevant to redistribute the oil. Where the redistributed oil goes to if it is not on the surface for intermediate deformation speeds? Which volume/depth is tested by ATM FT-IR?

2.)       Is there any physical meaning behind the used fitting function or is the chosen parable-like function completely empirical? Fitting four data points with a function having three free parameters cannot fail. Otherwise, it is questionable to discuss the obtained fit parameters. How trustable are the trends presented in Figure 9 considering this situation?

3.)       There is confusion with the labels in Figure 2. The labels “far”, “front” and “side” are used in the figure and in the lower part of the main text. In the sentence where the positions are introduced in the main text, however,  “far”, “front” and “near” are used. This is resulting in confusion and needs correction/clarification.

4.)       For me it makes not much sense to show so many quite similar IR spectra in 16 different plots. These curves could be given in much more compact way saving a lot of space without any loss of information. Negative values for the absorbance need an explanation.

5.)    There is not even one example showing experimental data which have been used for the determination of the tearing energy. Instead, technical drawings of the used specimen are shown. A plot showing the tensile data analysis would help the reader a lot to understand the lengthy described data evaluation method and to judge the data quality. The presentation strategy is quite different for original data from FT-IR and tensile measurements. It is hard to understand why.

6.)    The first sentence of the Introduction, sentences like “This is to picturize the process and is completely different from the plot area to calculate the tearing energy.“ or those telling why something is not shown should be deleted.

Minor points

a)       The abstract is completely written in “past”. This is unexpected from my point of view and will irritate the reader.

b)      The boxes in Figure 8 are useless and disturb reading.

c)       The English should be checked by a native English speaker. In some cases this produces even problems with understanding.

Author Response

Dear Reviewer,

Thank you for your valuable comments.

We authors are happy to implement major revision in the manuscript, abiding by most of the comments put forward by you.

Answers to the comments are attached as a word document named as "comments_answers_reviewer_1".

we believe that the manuscript appears to be much more complete after implementing the revisions.

Thank you indeed once again, and we wish you a nice day.

Regards,

Authors

Reviewer 2 Report

Review on “Parametrical Function Describing Influence of Redistribution of Incorporated Oil for Rupture Process Reconstruction in Rubber”

The authors describe a new (at least for me) and creative approach to assess the load conditions before fracture in technical rubbers. The idea is to exploit the stress/pressure-induced migration of the oil by post-mortem investigation of the oil’s absorbance peaks in the fracture zone, where supposedly the highest pressures were achieved.

However, language needs minor revisions. Even though I am not a native speaker I recognized numerous spelling, typesetting and grammatically errors, to name a few:

·         Line 30: initial -> initially

·         Line 33: were. further

·         Line 40: Reverse engineer a fractographic process? Fractography, as I understand it, IS the science of measuring and describing fracture processes.

·         Line 255: developed Thus

Regarding the introduction:

·         I think it is worth mentioning between lines 53 and 57 that the physical compatibility, e.g. quantified via popular chi or delta parameters, is certainly of utmost importance for miscibility and migration.

·         In CB filled NR, SIC onset depends on CB content and does not start universally at 200 %

The idea itself is interesting, because it is a major challenge to figure out the exact load conditions in the fracture zone. However, I have some major concerns regarding the overall reasoning and data evaluation:

·         First: Did I understand correctly that you measure IR in the fracture zone with the measurement head placed upon the rough, fractured rubber surface?

·         I cannot understand your conclusion that Table 2 supports the hypothesis that oil is squeezed out of the notch. The “side” is, according to Fig.2, the region of highest strain and stress, where hydrostatic pressure is presumably highest. However, results in Tab. 2 indicate largest IR peaks for the oil.

·         Statement Lines 248+249: At -195.7 °C the oil will hardly move at all, why then do this experiment?

·         Statement Line 257-259: You argue that at 10 mm/min the oil equilibrates and concentration is the same over the whole sample. However, you have previously shown that even in a static (“speed 0 mm/min”) experiment there is a gradient in concentration?

·         The most important issue I have is that you assume that oil migration is that fast to reflect the gradient in hydrostatic pressure, but on the other hand assume that it stays in the vicinity of the crack after stress release (rupture) sufficiently long to obtain your IR spectra. If you assume that it is squeezed out in terms of “leaving the rubber”, i.e. making a film on the ruptured surface, this should be clearly stated and supported by an experiment like measuring again after cleaning the surface with appropriate measures.

·         I am not sure how trustworthy and useful the fittings and “extrapolations” are, given the previous concerns.

Overall, I recommend major revision before publication. In case I understood something wrong I’d be glad to learn.

The topic itself is of technical relevance, as finite element calculation of rubber articles need to be more and more precise to approach design boundaries. However, it is not clear how the present articles presents significant advantage over models already present, see e.g. Plagge, J., et al., International Journal of Engineering Science 151 (2020): 103291 as a starting point.

The authors point out that they are probably the first to model recovery of (what they call) Mullins’ effect, but do not take into account the physical quantities defining recovery, e.g. temperature and time “at rest” and eventually remaining strains. Like this, the modelling resembles a mere curve-fitting problem with abundant parameters. With the given amount of parameters it is not an achievement to get precise fits like shown in the figures.

Moreover, Harwood and Payne [21], whose experiments are used for fitting, deal with unfilled (!) rubbers, i.e. pure natural rubbers without and fillers like carbon black or similar. Mullins’ effect is commonly attributed to be filler-induced and results in different ! The authors even write about this in line 33. The data from Harwood and Payne is most probably a strain-crystallization, protein-breakdown or polymer network issue. Even though the authors do not claim to build a physical model, they should get their wording right or choose appropriate experiments (which are out there in the literature).

Regarding the continuum mechanics behind the model, I do not have the time to dig into the details. From a bird’s eye, it looks reasonable but with too many arbitrarily chosen functions and parameters for the relative simplicity of the problem (given the fact that neither time nor temperature somehow enters).

Altogether, I strongly recommend major revisions before publication.

Author Response

Dear Reviewer,

Thank you for your valuable comments.

We authors are happy to implement major revision in the manuscript, abiding by most of the comments put forward by you.

Answers to the comments are attached as a word document named as "comments_answers_reviewer_2".

we believe that the manuscript appears to be much more complete after implementing the revisions.

Thank you indeed once again, and we wish you a nice day.

Regards,

Authors

Reviewer 3 Report

Please, see the attached document.

Author Response

Dear Reviewer,

Thank you for your valuable comments.

We authors are happy to implement major revision in the manuscript, abiding by most of the comments put forward by you.

Answers to the comments are attached as a word document named as "comments_answers_reviewer_3".

we believe that the manuscript appears to be much more complete after implementing the revisions.

Thank you indeed once again, and we wish you a nice day.

Regards,

Authors

Reviewer 4 Report

This paper requires substantial revision for publication.

1, The position of this study is not explained in the Introduction. Comparison with the Author's previous research is meaningless in this case.
2, The Experimental section needs to be revised for better clarity; not all readers of Polymers have sufficient knowledge of rubber.
3, If you want to show the concentration of oil in Figs. 3, Authors should improve the way you draw the figures.
4, The fitting of Figures 8 and 9 is problematic.

Author Response

Dear Reviewer,

Thank you for your valuable comments.

We authors are happy to implement major revision in the manuscript, abiding by most of the comments put forward by you.

Answers to the comments are attached as a word document named as "comments_answers_reviewer_4".

we believe that the manuscript appears to be much more complete after implementing the revisions.

Thank you indeed once again, and we wish you a nice day.

Regards,

Authors

Round 2

Reviewer 1 Report

Parametrical Function Describing Influence of Redistribution of Incorporated Oil for Rupture Process Reconstruction in Rubber by Sanjoy Datta, Radek Stoček and Evghenii Harea.

The authors have improved the quality of their manuscript and incorporated additional parts which help to understand why the study has been performed, how IR measurements have been done and what is the relevance of the main conclusions drawn here. This underlines that the experiments are carefully done and why the results are interesting. Although I am still a bit sceptical since there are far-reaching claims based on limited experimental data it is now at least clearly mentioned in the manuscript what is really done. Hence, I can recommend publication of the manuscript in Polymers without having bigger concerns. A main argument for publication of the manuscript is that a novel and interesting approach to describe the interrelation between crack propagation and oil distribution is proposed.

Author Response

Response to reviewer 1

Parametrical Function Describing Influence of Redistribution of Incorporated Oil for Rupture Process Reconstruction in Rubber by Sanjoy Datta, Radek Stoček and Evghenii Harea.

Comment: The authors have improved the quality of their manuscript and incorporated additional parts which help to understand why the study has been performed, how IR measurements have been done and what is the relevance of the main conclusions drawn here. This underlines that the experiments are carefully done and why the results are interesting. Although I am still a bit sceptical since there are far-reaching claims based on limited experimental data it is now at least clearly mentioned in the manuscript what is really done. Hence, I can recommend publication of the manuscript in Polymers without having bigger concerns. A main argument for publication of the manuscript is that a novel and interesting approach to describe the interrelation between crack propagation and oil distribution is proposed.

Response to the comments: Thank you for your valuable as well as informative and hopeful comments. We would say that the work was an initial attempt to find a trend between the speed of deformation to total rupture in a uniaxial tensile deformation speed programme and the redistribution of the initially homogeneously incorporated oil. We agree that we performed limited experiments and with those, though a trend was found but the work can be definitely improved upon in future with a greater number of relevant experiments. We acknowledge your great effort in the first review which opened up an opportunity to greatly improve upon the first manuscript. Thank you indeed once again

Reviewer 2 Report

The major issues were resolved appropriately. So I suggest publication.

I have to apologize to the authors: Obviously, there was a copy-past mistake in my review report such that it contained two reviews, the second being for another modeling-centered work. Of course, these comments can be (could have been) ignored. Sorry!

Author Response

Response to reviewer 2

Comments and Suggestions for Authors: The major issues were resolved appropriately. So I suggest publication.

I have to apologize to the authors: Obviously, there was a copy-past mistake in my review report such that it contained two reviews, the second being for another modeling-centered work. Of course, these comments can be (could have been) ignored. Sorry!

Response to the comments: Thank you for your valuable as well as informative and hopeful comments. We would say that the work was an initial attempt to find a trend between the speed of deformation to total rupture in a uniaxial tensile deformation speed programme and the redistribution of the initially homogeneously incorporated oil. We agree that we performed limited experiments and with those, though a trend was found but the work can be definitely improved upon in future with a greater number of relevant experiments. We acknowledge your great effort in the first review which opened up an opportunity to greatly improve upon the first manuscript. Thank you indeed once again

Reviewer 3 Report

This manuscript needs a serious revision of English language. Shorter sentences are better than long unintelligible phrasing.

Author Response

Response to reviewer 3

Comments and Suggestions for Authors: This manuscript needs a serious revision of English language. Shorter sentences are better than long unintelligible phrasing.

Response to the comments: Thank you for your valuable omments. We would say that the work was an initial attempt to find a trend between the speed of deformation to total rupture in a uniaxial tensile deformation speed programme and the redistribution of the initially homogeneously incorporated oil. We have performed limited experiments and with those, though a trend was found but the work can be definitely improved upon in future with a greater number of relevant experiments. We acknowledge your great effort in the first review which opened up an opportunity to greatly improve upon the first manuscript.

As for the English language, thank you thank you indeed for your suggestion. It is advisable to write short sentences over writing a long one. Abiding by your suggestion, we have now reshaped the entire manuscript accordingly.

Reviewer 4 Report

The first half of the paper has been improved by corrections to the Reviewer's comments. On the other hand, there are still some problems with the article. For example, Figure 3 shows a plot of Deformation versus Force, which is not consistent with Table 2, for example. The original plot of Deformation and Force is not appropriate for a scientific paper. There are also problems with Figures 9 and 10; for Figure 9, the authors should provide a clear rationale for the second order of the approximation equation. As for Figure 10, it is too crude an approximation for the number of data; if it is to be shown, the slope should be displayed.

Author Response

Response to reviewer 4

Comments and Suggestions for Authors: The first half of the paper has been improved by corrections to the Reviewer's comments. On the other hand, there are still some problems with the article. For example, Figure 3 shows a plot of Deformation versus Force, which is not consistent with Table 2, for example. The original plot of Deformation and Force is not appropriate for a scientific paper. There are also problems with Figures 9 and 10; for Figure 9, the authors should provide a clear rationale for the second order of the approximation equation. As for Figure 10, it is too crude an approximation for the number of data; if it is to be shown, the slope should be displayed.

Response to the comments: Thank you for your valuable suggestion. Related to the confusion and the mismatch arising between Table 2 and Figure 3, we now try to clarify the confusion. Table 2 shows the redistributed oil concentration for various static deformation programmes. These static deformation experiments were done separately and also before doing the dynamic deformation experiments. In the manuscript, Figure 3 shows the calculation of the tearing energy under the dynamic condition. In the experimental part we have clearly described that we did two different types of experiments, one under the static condition and the other under the dynamic condition.

Related to Figure 3, we appreciate that a more scientific way to represent it is to show the data in stress-strain coordinates. However, we decided to plot the figure arresting the information in the force-displacement coordinates, reasoned out to the fact that the calculation of the tearing energy would appear to be clearer to the readers in this form. This figure was to show how the calculation of the tearing energy was done by the “absolute area type integration”.

Plotting of Figure 9 was done on a first trial basis. Of course, the acquiring of more results in the future will open up the possibility of fitting of data in equations in a better manner – more related to reality. At present, abiding by your valuable comments, we have now grossly modified the manuscript by describing the benefits and limitations of Figure 9.

As for Figure 10, the aim of this work was not to exactly quantify the concentration of the redistributed oi under a defined deformation programme but to find a trend between the initial oil concentration and the redistributed oil concentration for various tensile ruptured programmes. This does not claim by any means a very high degree of accuracy but a general trend in the observed results. The slopes of all the straight lines in the shown in Figure 10 are mentioned in Table 5. This was done to avoid crowding in the figure area.

In general, we have grossly modified the entire manuscript to make it more comprehensible

Round 3

Reviewer 4 Report

The authors have appropriately addressed the comments.